# Phosphorus Dynamics in the Soil–Plant–Environment Relationship in Cropping Systems: A Review

**Rodolfo Lizcano-Toledo** [1,2]**, Marino Pedro Reyes-Martín** [1,]*** , Luisella Celi** [2] **and Emilia Fernández-Ondoño** [1]

[1] Department of Soil Science and Agricultural Chemistry, University of Granada, Av. de Fuente Nueva s/n, 18071 Granada, Spain; rodolfolizcano@correo.ugr.es (R.L.-T.); efernand@ugr.es (E.F.-O.)

[2] Department of Agricultural, Forestry and Agrifood Sciences (DISAFA), University of Turin, Largo Paolo Braccini, Grugliasco, 10095 Torino, Italy; luisella.celi@unito.it

[*] Correspondence: marinoreyes@ugr.es; Tel.: +34-645649346

**Abstract:** This work performs a review of the relevant aspects of agronomic dynamics of phosphorus (P) in the soil–plant relationship as a community (crop ecophysiology), the effect of environmental conditions and global warming on the redistribution and translocation of P in some crop, and the use of good agricultural practices with the aim of improving the efficiency of the element. The research focuses on Northern Europe, North-Eastern Asia, Oceania, North America, and the tropical area of Latin America. This review covers general research and specific works on P found in the literature, 70% of which date from the last 10 years, as well as some older studies that have been of great relevance as references and starting points for more recent investigations. The dynamics of P in a system implies taking into account genetic aspects of the plant, component of the soil–plant–fertilizer–environment relationship, and use of technologies at the molecular level. In addition, in a climate change scenario, the availability of this element can significantly change depending on whether it is labile or non-labile.

**Keywords:** fertilizers; struvite; agricultural practices; environmental impact





## 1. Introduction

Phosphorus is a key macronutrient for the growth and development of crops. It participates in the synthesis of phospholipids and nucleotides, which are structural components of cells and their membranes. It is also involved in membrane permeability, photosynthesis, respiration, glycolysis, redox reactions, and signal transductions (mechanisms that enable plants to obtain information from the environment and adapt themselves to its constant changes). In short, a response between the cells and the growth regulators requires receiving proteins, such as the plant–fungus interaction signal, which occurs especially under P-deficiency conditions [1]. Additionally, P participates in lipid metabolism, carbohydrate transportation, and the maintenance of osmotic potential [2–4]. On the other hand, a reduction of bioavailable P causes changes in its relationships with other elements (stoichiometric relationships) such as carbon (C) and, especially, nitrogen (N) [5–7].

As a dynamic element in nature, P is vital in agriculture, where it is a basic fertilizer [8]. The P dynamics depend on soil factors such as pH, salinity [9], high concentrations of toxic elements, interaction with micronutrients [10], runoff, leaching [11], total P concentration, soluble (and, as such, assimilable for the plants) P pool [12], organic matter content, redox potential [13], soil structure and texture (especially the clay content) [14,15], mineralogical composition [12], enzymatic activity, P-solubilizing microorganisms [16,17], and mycorrhizas [18].

Low available/labile P concentrations in the soil require high applications of P fertilizers, although this practice can pose serious environmental issues [19]. Recent studies have identified the so-called agricultural diffuse pollution of P [20], that is, water pollution arising from a broad array of human activities for which the pollutants have no obvious

point of entry into receiving watercourses [21]. This pollution comes mainly from crop fertilizers applied to the soil [22,23].

Crops absorb P from the soil through the diffusion process, but the diffusion coefficients of this element is very low and its concentration in the soil solution is limited [24–26]. Hence, phosphatized fertilization (due to the low mobility of this element in the soil) must be applied in the proximity of plant roots, either in bands (especially six-month crops) and/or half or full circle [27].

Within the plant, P is highly mobile, as opposed to its activity in soil [28]. P deficiency causes a characteristic coloring ranging from orange to reddish tones in the old leaves (Figure 1). This is due to a reduction of the chlorophyll biosynthesis and to a higher production of pigments such as anthocyanins [29,30]. Soluble P is transported through the xylem to all growth points, depending on P concentration, which is located within the adequate range of 0.1–0.3 g P kg$^{-1}$ for most crops (grasses and agro-industrial crops, including fruit, legumes, and others) [31].

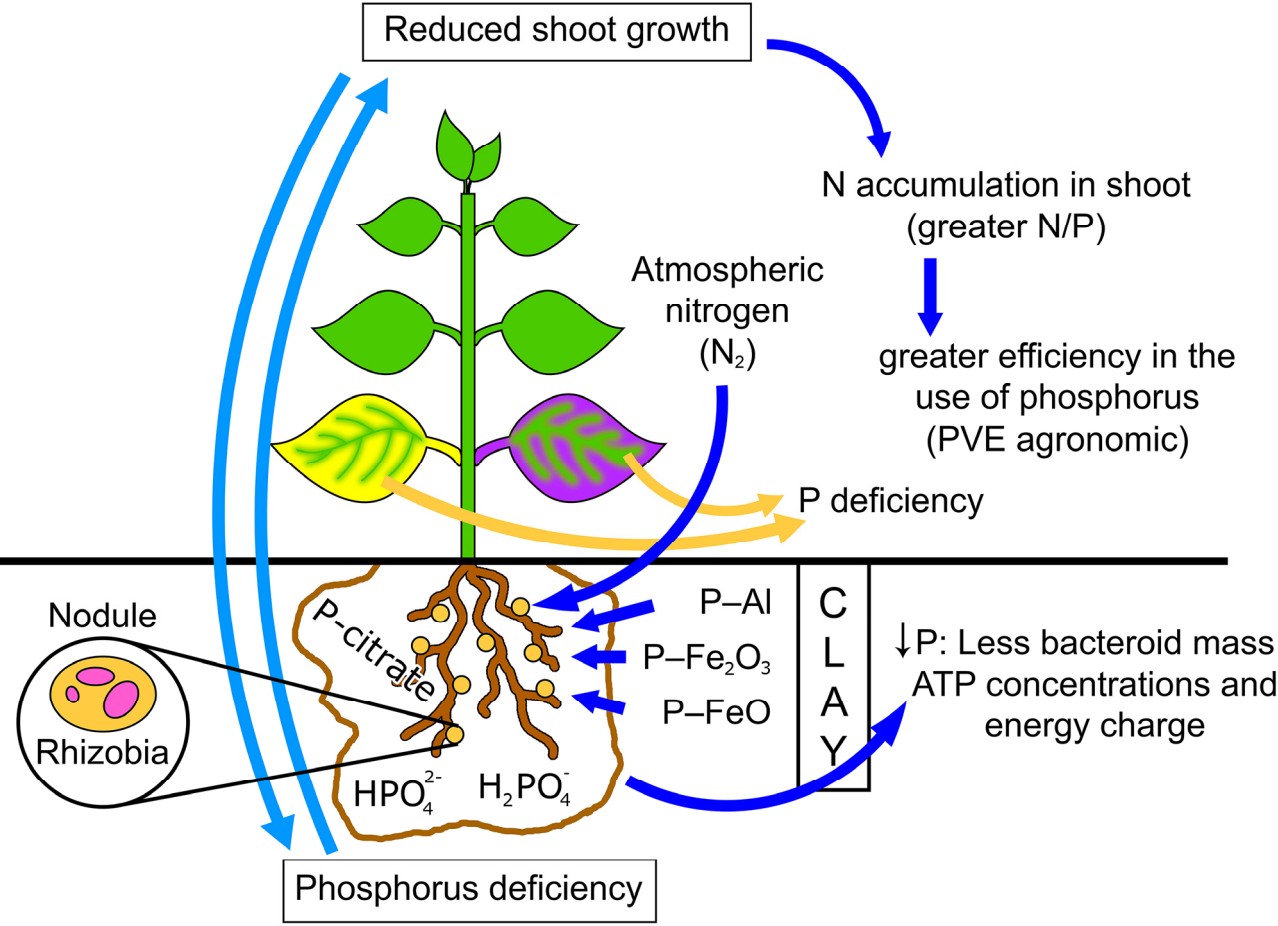

**Figure 1.** Phosphorus cycle and N fixation process in the plant–soil–environment system.

Although P is important for crops, its dynamics are involved in relationships with other elements. Since P is a macronutrient that limits primary production in many ecosystems, the removal of C from the atmosphere may require large amounts of P. Spohn [32] showed that the storage of organic C in mineral soils sequesters large amounts of P. The reasons for the strong enrichment of organic P in soils are mineralization of organic C and formation of microbial necromass rich in P, as well as the strong sorption of organic P to mineral surfaces that prevent its mineralization.

The deficiency of supply and availability of phosphorus remains a severe limitation in biological nitrogen fixation and symbiotic interactions. This requirement may be higher than for the growth of roots or shoots of the host plant. There are marked differences in the requirements of plants and rhizobia for P [33], with slow-growing ones more tolerant to low levels of P than fast-growing rhizobia [34]. The nodules themselves are strong sinks of P [35], and nodulation and $N_2$ fixation are strongly influenced by the availability of P [36].

The phosphatized fertilizers most used for agriculture are water soluble and neutral ammonium citrate soluble or citric acid soluble phosphates [37–39]. Simple superphosphates (SSP), triple superphosphates (TSP), monoammonium phosphate (MAP), and diammonium phosphate (DAP), apart from being used in traditional ways, are the most soluble P sources and the ones that provide the highest efficiency for crops, although they are also the ones that contribute most to environmental pollution.

These fertilizers are costly, as their production demands require a high amount of energy, apart from requiring other elements to be soluble, such as sulphuric and nitric acid (imported raw material, which increases the selling price of the product even more). Hence, research is needed on phosphatized controlled-release fertilizers which can minimize the losses outside the crop in the environment and improve the agronomical effectiveness of the fertilizer use [40,41]. In this context, materials such as biochar are being tested [42], or others with water-soluble P coatings with an insoluble layer, which creates a physical barrier to slow down the release rate of P. The polymers applied include graphene oxide, a nanomaterial which is being studied as a middle-term P carrier for the crops [43,44].

This review seeks to explain P dynamics in an agronomic context, evaluating the soil, plant, and environment nexus, while assessing the effects of climate change on P availability in cropping systems and identifying agricultural practices that aim to find the best trade-off between crop productivity and environmental sustainability.

## 2. Materials and Methods

### 2.1. Selection of Search Criteria

2.1.1. Recent Results: Starting When? When Will We Apply Results from Other Years?

This review compiles information regarding research on P from the literature, 70% of which dates from the last 10 years, as well as some older studies that are considered a reference and a starting point for recent investigations.

The search terms were carried out according to the following entries: P in different types of environments, oxide reduction processes, role of Phosphorus in the biological fixation of N, conventional and alternative sources of P in agriculture, effect of climate change in soil availability, stoichiometric ratio C:N:P, and dynamics of P in the soil–plant relationship; we also looked for peer-reviewed studies that report on the use of 32P isotope in agriculture.

2.1.2. Priority Geographical Location

This review focuses on Europe (20%), Asia (31%), Oceania (10%), North America (15%), South America (11%), and other regions (13%). However, in this study on P dynamics, no difference is made between these locations.

2.1.3. Agronomic Criteria

The following agronomic criteria were selected: (1) soil P processes (adsorption, leaching, precipitation, loss to surface runoff, immobilization due to microorganisms); (2) P as a

nutrient for plant (vegetal physiology) and for crop (community physiology), to determine the characteristics of nutritional P deficiency in various crops and their impact on productivity and quality; (3) the role of microorganisms on driving P availability for the crops; (4) P dynamics in the environmental processes and their impact on yield in major crops worldwide; (5) the effects of P on the biogeochemical processes in relation to global warming; (6) the agronomic efficiency for conventional and alternative phosphatized fertilizers aimed at increasing productivity and decreasing environmental impact in agricultural systems.

### 2.1.4. Databases

The literature search was made primarily in the ISI Web of Knowledge database, combined with Science Direct. The database was selected due to the comprehensive content of journals and articles relevant to the subject. A search was carried out for each of the combinations of criteria (n = 40), and articles were selected based on their relevance to the topic location within NW Europe, and soil studies in the American tropics, China, and Australia (n = 110).

### 3. Characteristics of Soil P Pool Processes in the Soil

Losses and limitations of P in soil are quite high as, on the one hand, low-solubility fertilizers (Thomas slag and phosphoric rock) and high-solubility fertilizers can be used, among which there are MAP, DAP, TSP, and $H_3PO_4$ used in fertigation programs. On the other hand, processes such as adsorption, precipitation, and microbial immobilization can occur due to physico-chemical and microbial conditions of the soil [45].

Phosphorus can be found in two different forms in soil, either organic or inorganic, presenting a continuous lability and availability in the soil solution [46]. Phosphate sorption is determined mainly by iron and aluminum oxyhydroxides and occurs mainly within the low-crystallinity forms and with positive charges. This adsorption occurs at Lewis acid sites, where the -OH and $-OH_2^+$ groups, mono- or tricoordinatedly bound to the metal (Fe and Al), are exchanged by the phosphate [47]. Thus, the equilibrium between adsorption and desorption is mainly related to soil pH [48]. Depending on the pH, P exists in the soil solution as orthophosphates ($H_2PO_4^-$ and $HPO_4^{2-}$). According to [9], the soils of the Yellow River delta in China showed a maximum P sorption rate (43.9 mg kg$^{-1}$, 45.9 mg kg$^{-1}$, and 40.5 mg kg$^{-1}$ in different types of soils) with optimal pH levels of 5.5, 6.2, and 5.2, respectively.

According to [48], P sorption reaches its maximum value in the pH range between 5.0 and 7.0. This was assumed to be related to the presence of soluble forms of P, such as $H_2PO_4^-$ or $HPO_4^{2-}$. combined with a positive charge of pH-dependent surfaces.

At high pH values, P can precipitate with Ca, forming amorphous calcium phosphates, octocalcium phosphates, and apatites (hydroxyapatites or fluorapatites). This is again pH dependent, increasing as the pH increases [49].

Crop productivity has been estimated to decrease by as much as 40% with marked deficiencies of P in the soil [50], in Asia [51,52], America [53,54], Europe [55], Africa [56], and Oceania [57].

The P release in agricultural soils represents an important threat for water quality [58]. P losses can be due to runoff, leaching, and erosion in soluble forms (<0.45 mm) and particles (>0.45 mm). Normally, P particles are the main nutrient transported from cropped soils to waters, representing 80% of all carriers materials [59,60]. This is because of the selective erosion of P-rich particles caused by rainfall, splash, and cutting forces of water flow over the land surface according to the topography [61,62]. On the other hand, losses resulting from surface runoff (in dissolved particles and forms), subsurface flow (leaching and flow through the soil matrix and macropores), drainage flow, or even from groundwater can boost the eutrophication of the water [63,64].

Various studies have confirmed that this high risk of surface-water pollution arising from processes such as runoff during periods of high precipitations (rainfall regime) is linked to the high solubility of these materials [65–67]. Authors [68] found out that the P

loss due to runoff was similar for different soluble mineral fertilizers such as MAP, DAP, and $KH_2PO_4$.

The high fixation capacity of P in most soils in the world and the low efficiency of the use of P-based fertilizers (around 10–15%) in most crops mean that an excess of P entry tends to accumulate in soils [69]. This accumulation can become a problem due to the different types of pollution it can generate, as mentioned above, and to the high cost of the phosphatized fertilizers.

In Europe and Oceania, for instance, there is an accumulation of phosphatized fertilizers input in agricultural soils (ranging between 560 and 1.115 kg P ha$^{-1}$, respectively, for the period 1965–2007). We witnessed a dramatic increase of phosphatized fertilizers input in agricultural soils in Europe (560 kg P ha$^{-1}$) and Oceania (1115 kg P ha$^{-1}$) for the period 1965–2007, well above the amount absorbed by the corresponding crops, and these were much higher than the absorption accumulated by the crops (100 kg P ha$^{-1}$ in Europe and 350 kg P ha$^{-1}$ in Oceania, respectively) [70]. At the same time, this leads to changes in the soluble P dynamics in the soil solution pools, in the microbial and enzymatic activity of the rhizosphere zone, and in the absorption, extraction, and transportation of P into the plant.

## 4. Phosphorus Dynamics in the Plant

Phosphorus is an irreplaceable and essential element for life. In plants, it is one of the 16 essential elements for their vital functions, and thus without this element, no high-quality or abundant harvests could be achieved. Nevertheless, the biomass potential in soil is limited to the low P reserves worldwide [71].

As a macronutrient for plants, P is fundamental to the development of many cellular components, such as the nucleic acids, proteins, phospholipids, and ATP [72]. It is essential for the enzymatic regulation and for the interpretation of metabolic signals, and its important mobility within the plant causes the signs of their nutritional deficiency to be reflected first in the old leaves [73,74], although recent works [75] point out that plants can be heavily affected by a P deficiency for weeks without reflecting any visual symptoms on the leaves. On the other hand, all photosynthetic processes that are influenced by P deficiency seem to be fully reversible and can be restored in less than 60 min by providing orthophosphate to the leaf tissue.

To grow under P deficiency, plants have developed several strategies [76] that can be classified into two groups: (i) those improving P absorption from soil, and (ii) those limiting P reallocation in vegetal organs [77]. To increase P uptake from soil, plants may modulate their root system architecture by increasing their surface area and producing different types of special roots called proteoid roots or radicles and basal roots [77–79]. This leads to the exploration of larger soil volumes, scavenging for P, and to modification of rhizosphere environment to boost P mobilization. Roots may indeed increase the release of exudates, acidifying the rhizospheric area to produce more $H_2PO_4{}^-$, which is the preferential ion for acquired roots. The exudation includes also the release of organic acids that can compete with P for the same sorption sites, and enzymes, such as phosphatases and phytases that can hydrolyze different organic P forms [80–82]. The associations of plant roots with mycorrhiza, especially arbuscular mycorrhiza, may further improve the capacity of plants to acquire P via the hypha network that increases the spatial exploration [83,84].

By contrast, plants limit the use of P as their growth rates decrease, by accumulating sugars and anthocyanins, modifying their glycolytic and respiratory pathways, and by alternating vegetal hormones [85,86].

Other special strategies that plants have developed to take up P include the capacity to produce different types of special roots called proteoid roots or radicles and basal roots. The second one is the development of associations with mycorrhizas, especially arbuscular mycorrhiza, forming a symbiosis with the plant roots, so the hyphas of these fungi increase the spatial availability of P. The third strategy consists of modifying the environment of the rhizosphere in order to boost P mobilization. This includes the release of exudates by the plant roots, acidifying the rhizospheric area so there is more $H_2PO_4^-$, which is the

easier form that plants can absorb (physiological factor) that is found only in acid soils. The exudation of the rhizosphere also boosts the production of phosphatases, as it mineralizes the different forms of organic P in the soil.

Plants produce many secondary metabolites that are fundamental for survival under environmental stress. These include flavonoids and especially anthocyanins whose biosynthesis is modulated not only by a number of intracellular signals but also by environmental stimuli [87,88]. The problems regarding P deficiency in crops cause anthocyanins to accumulate in plants.

These adaptive responses are due to protein production, arising from the differential expression of many stress-sensitive genes [89]. Therefore, identifying the genes and/or proteins involved in the adaptive responses to P deficiency is essential in order to understand the molecular mechanisms of the adaption. A strategy such as the accumulation of anthocyanins in the shoot is related to proteomic studies. Authors [29], on the basis of the iTRAQ-based proteomic analysis with *Arabidopsis,* determined the molecular mechanisms for the anthocyanin accumulation in induced plants under P suppression. The authors concluded that, under P deficiency, this flavonoid increased its metabolite concentration, with high levels of mRNA of seven proteins mapped in the experiment. This suggests that P deficiency promotes the accumulation of anthocyanins by prompting their biosynthesis. Hence, when plants undergo early deficiencies of P, the mature leaves (due to the high mobility of P within the plant) present violet, orange, or red tones due to the increase of anthocyanin concentration around the leaf area, with a remarkable decrease of photosynthesis.

## 5. Role of Soil Microbiota on P Processes

Sustainable agriculture is the main strategy to fight the quick deterioration of the environmental quality by maintaining the ecosystem balance. Some restrictions can limit crop productivity, such as limited P in the soil, which frequently needs to be restored repeatedly in order to satisfy the demand of the plant. This is partly due to its quick loss when in its labile form, arising from the extraction of the element (P) by the crop (nutritional requirement). Furthermore, other processes include adsorption, precipitation, and microbial immobilization.

The tight interaction of microorganisms with roots may provide a greater availability of nutrients for plant (Figure 2). Some microorganisms have the ability to promote plant growth and productivity and are recognized as plant-growth-promoting microorganisms (PGPM) [90]. Phosphate-solubilizing microorganisms (PSB), for example, constitute an important group of PGPM, since they are involved in a wide range of processes that affect P transformation, being integral components of the soil cycle of this nutrient [91].

Bacteria are the predominant microorganisms that solubilize mineral phosphate in soil when compared to fungi and actinomycetes [92,93]. In general, solubilizer bacteria outnumber fungi from 2- to 150-fold [94,95]. On the other hand, most solubilizing microorganisms can solubilize calcium phosphate complexes and only some of them can solubilize aluminum or iron phosphates [96].

As plants, microorganisms have developed several mechanisms to solubilize and mobilize P, including the release of P-solubilizing protons organic acid anions such as oxalate or citrate that solubilize inorganic P by chelating $Al^{3+}$ and basic cations ($Ca^{2+}$, $Mg^{2+}$) bound to P, siderophores and polyphenols, and extracellular phosphatases that hydrolyze organic P [97–99].

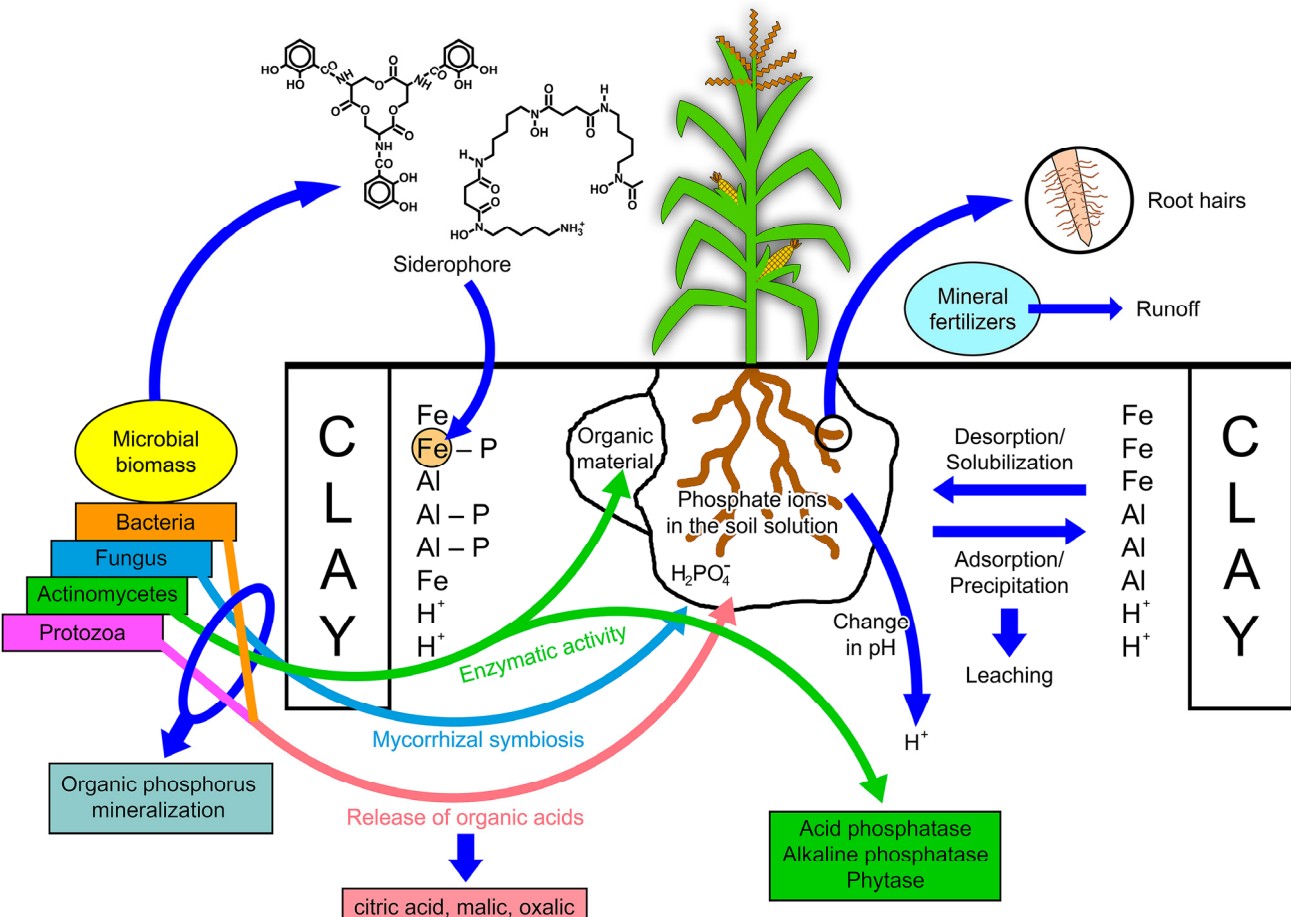

**Figure 2.** Dynamics of some microbial agents that present a high activity rate in the worldwide agricultural systems.

While microorganisms are able to produce both acid and alkaline phosphatases, plants can produce only acid ones [100–102].

In addition, microorganisms directly affect the ability of plants to obtain P from the soil by increasing the extension of the root surface area through mycorrhiza associations or by promoting the growth of lateral roots and root-absorbing hairs through phytohormones [103].

Microorganisms may affect P bioavailability due to the tight link with C and N cycles. Microbial organic matter mineralization generates a stoichiometric C:N:P relationship that must be considered in the application of fertilizers, either using N and P fertilizers combined with organic matrices, or maintaining crop residues with direct seeding or reduced tillage management [104]. Some studies [105] observed that applying N and P fertilizers increased rice straw mineralization by 25% and 10%, respectively. This is because N and P addition stimulates microbial activity, with higher $CO_2$ flow but a lower emission of methane. In addition, a combination of fertilization with N and P stimulated microbial activity, resulting in a higher production rate of extracellular enzymes, which accelerated organic matter decomposition. On the other hand, the high $CO_2$ emission under P deficiency depressed rice productivity in terms of biomass. Some studies [106] determined that the response of $CO_2$ to low P concentrations ($<1.8$ g kg$^{-1}$ in the leaf) was higher than with greater concentrations. This may be related to diverse P-recycling phases within the rice plant, which has multiple stems depending on phenology stage, and reflects the high P mobility and its redistribution among different organs or phonologic structures [107].

## 6. Phosphorus Dynamics and Effects of Climate Change

Different environmental factors modify P dynamics in the soil–plant system, such as temperature, water stress, wet/dry cycles, drought, and $O_2/CO_2$ concentration [108]. These conditions can affect the dynamics in the absorption, transportation, and distribution of P from soil into the plant with contrasting effects.

Global change involves important changes on these factors with important impacts on P cycling [105].

The high $CO_2$ concentration reached in the atmosphere [109] anticipates an improvement in the photosynthetic rate of the crops, and some physiological variables, such as the Net Assimilation Rate, which reflects $CO_2$, can act as a fertilizer for the crops, contributing to the net primary productivity [110,111]. Nevertheless, an increase in the plant growth causes a higher nutrient demand, leading P to become the most limiting nutrient [112].

The worldwide demand for fertilizers in 2017–2018 was 187 million tons of P, and this number is expected to increase up to 199 million tons for 2022–2023 due to demographic growth, which demands a higher productivity [113]. The indiscriminate use of phosphatized fertilizers not only raises production costs, but also generates massive environmental pollution and a major reserve of P in agricultural soils [114]. In general, the response of fertilizers in the irrigated areas, especially the areas used to grow rice throughout the world, decreased around two-thirds, from 13.4 kg grain per NPK kg in 1970 to 3.7 kg grain per NPK kg in 2005 [115]. These fertilizers, due to their alkaline nature, also affect the dynamics of enzymes and bacteria that solubilize P. This is due to the alkaline-residual effect that those fertilizers present when entering in contact with water and dissolving in the soil, which leads also to changes in the microbial respiration [105].

In addition, an indiscriminate use of P fertilizers such as MAP (12-52-0) and DAP (18-46-0), when applied in productive agricultural systems, generates a high amount of nitrous oxide ($N_2O$), which is one of the most dangerous greenhouse gases, with a warming potential 298 times higher than that of $CO_2$ [116]. The Colombian Institute for Hydrology, Meteorology, and Environmental Studies reported that in 2004, the agricultural industry in Colombia produced 94.91 gigagrams (Gg) of $N_2O$.

There is an uneven distribution of high precipitations in different parts of the world, and the fact that 70% of soils worldwide are acid [117] should be also taken into account when applying phosphatized fertilizers acidulated with $H_2SO_4$, $HNO_3$, or $H_3PO_4$, which can further intensify soil acidity and hence P precipitation due to increasing amount of soluble Al and Fe [118,119].

Table 1 shows some worldwide research on the main processes in which P intervenes in agricultural systems.

**Table 1.** Dynamics of the procedures related to P in crops.

| P Forms | Area | Factors | P Activation Response on Soil or Plant | Crops | Reference |
|---|---|---|---|---|---|
| C:P:N | Stoichiometric relationships | Physiological, cellular, morphological, and molecular factors | | *Oryza sativa*<br>*Leymus chinensis*<br>*Panicum maximum*<br>*Bouteloua gracilis*<br>*Populus deltoides* | Dias Filho et al., 1992; Elser et al., 2010; Bell et al., 2014; Li et al., 2016; Shang et al., 2018; Zhu et al., 2018. |
| P deficiency—$H_2PO_4^-$ | Radicle and proteoid root emission | Physiological, cellular, morphological, and molecular factors | mRNA and ethylene synthesis | *Arabidiopsis* sp. | Ramaekers et al., 2010; Haling et al., 2013; Neumann, 2016. |
| | Alteration of biochemical processes | Biosynthesis of secondary metabolites, radicle emission (proteomic hairs) | Anthocyanines | *Arabidiopsis* sp. | Wang et al., 2018 |

**Table 1.** *Cont.*

| P Forms | Area | Factors | P Activation Response on Soil or Plant | Crops | Reference |
|---|---|---|---|---|---|
| Effect of global change on the P dynamics | P dynamics with temperature, water stress, greenhouse gases. | $CO_2$<br>$N_2O$<br>$O_3$ | Photosynthesis, metabolic expenditure, transpiration, microbial activity | *Zea maiz*<br>*Triticum vulgare*<br>*Orysa sativa* | Goufo et al., 2014<br>Wang et al., 2014<br>Hidayati et al., 2019 |
| Environmental impact of phosphatized fertilizers | Coating of the P molecule with different materials from different sources | Loss of soil, runoff, leaching, eutrophication | | *Zea maiz*<br>*Triticum vulgare* | Shaviv and Mikkelsen, 1993; Novoselov et al., 2012; Andelkovic et al., 2018 |

## 7. Efficiency of P Fertilizers in Agronomic Management and Environmental Impact

The agronomic efficiency of fertilizers is not yet adequate. This is partly due to the low nutrient-absorption rate in most soils and partly to losses [19]. Over 70% of the arable surface on Earth needs a great supply of P to produce crops. On the other hand, its absorption by plants is slow over short distances of approximately 0.1 to 0.15 mm [120]. According to other authors, the mobility and concentration of Phosphorus in soils is very low when compared to that of other nutrients; diffusion coefficients of phosphate in the soil of $0.3 - 3.3 \times 10^{-13}$ m$^2$ s$^{-1}$ are presented, and its concentration in the soil solution is 0.02 ppm [121].

The amount of fertilizer applied to a crop but not used by plants implies high environmental and economic costs. The environmental costs include pollution due to eutrophication and groundwater or runoff water hypoxia [122]. The P enrichment due to human activities is one of the main reasons and causes of the eutrophication of surface waters, which leads to an explosion in autotrophic organisms that can remarkably affect the quality of aquatic ecosystems [123].

This problem has led to the development of different types of technologies to obtain phosphatized fertilizers with a great agronomic efficiency in the soil, high yields, and low environmental impact.

The choice of the P source as fertilizer depends on its capacity to meet the needs of the crop plants during their phenological phases (agronomic efficiency) and on its solubility rate. Other aspects (quality traits of the fertilizers) can be considered when choosing P-based fertilizers, such as supply of other nutrients (synergy), pH, granulometry, mechanical resistance, segregation, density, hygroscopicity, saline index, and physical and chemical compatibility in the mixes. Figure 3 shows the agronomic efficiency of different sources of phosphatized fertilizers, the potential pollution, as well as new alternative sources of fertilizers in the worldwide market.

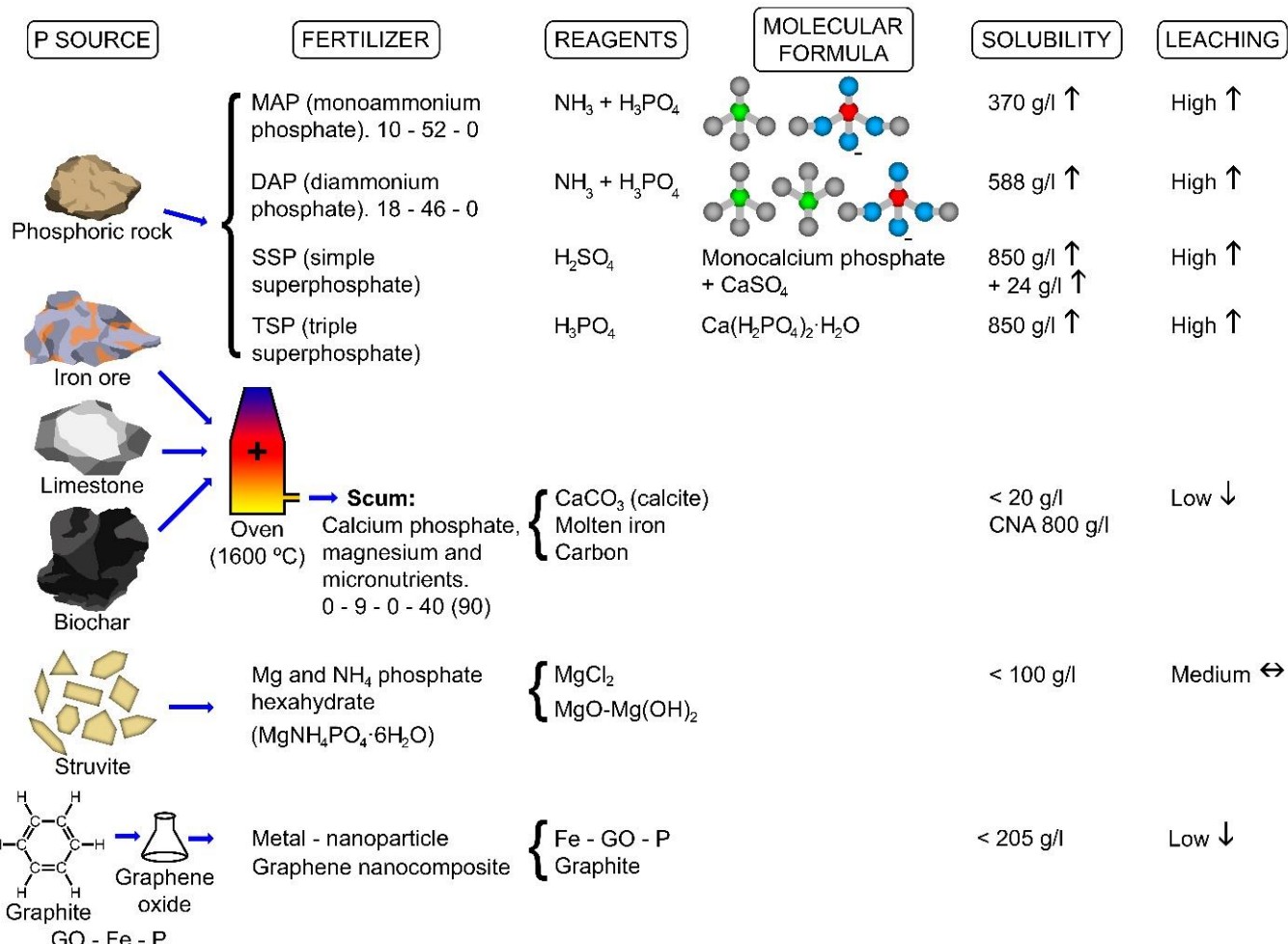

**Figure 3.** Different sources of phosphatized fertilizers in the market and their agronomical efficiency and environmental impact on the soil.

## 8. Alternative Sources of Phosphate Fertilizers of Potential Use in Agriculture

Of the total amount of P applied into the soils, only a small part can be recovered in the first harvest, and this leaves a remnant of potential residual P, which is not bioavailable and can be used by the plants during the following seasons [124,125]. This is an important reason for the attention devoted to developing phosphatized controlled-release fertilizers, as these can minimize the loss outside the application area and improve the agronomic use efficiency [40,41].

A new approach to face this challenge involves coating soluble P with an insoluble layer of different materials. This creates a physical barrier that can slowdown the release rate of P, reduce its leaching, and diminish other critical processes that degrade soils, such as the surface runoff. A number of natural and synthetic polymers have been studied, ranging from starch [126,127], cellulose [128,129], chitosan [129], and P acrylic acid acrylamide/kaolin [130].

Other types of materials have been tested as well, such as struvite ($MgNH_4PO_4 \cdot 6H_2O$), which is a mineral phosphate that can precipitate in wastewaters of treatment plants. In the last decade, struvite has gained interest as a method to recover P from wastewaters so it can be used as a source of P [131]. During the manipulation and recovery of this material, different processes may be applied, which can be classified into two groups. The first group uses magnesium (Mg), most frequently as $MgCl_2$, and the pH is adjusted by using NaOH. Magnesium in form of water-soluble salt reacts quickly with P present in the wastewaters during the crystallization process, thus enabling the formation of a highly pure product

(struvite). Authors [132] used alternatively MgO and Mg(OH)$_2$ as Mg source, which is more economical than any other Mg-based salt and limits the use of NaOH. These reactions are slower than those of other Mg salts, as they have to dissolve first, so this is normally added excessively in order to boost the formation of struvite, resulting into a product with an excess of MgO or Mg(OH)$_2$ [133,134].

Experimental tests have been also performed in different crops with products derived from metallurgy in order to decrease the use of phosphoric rocks. Metallurgic industries of Ta and Nb produce a waste of acids with an acidity of 5.8 mol L$^{-1}$, a high content of S [135], and a pH of around 0 [136]. This type of acid source has been used to solubilize low-reactivity apatites, involving a lower price and lower-solubility P.

The potential to improve the formulations of fertilizers by using nanomaterials has been explored as well [44]. Carbon-based materials are often applied in agriculture [137]. Their lower solubility makes them more profitable, environmentally friendly (less leachable), and chemically and physically advantageous.

One of these materials is biochar. The release of carbon dioxide as one of the main greenhouse gases is considered a key negative effect associated with the burning of agricultural waste by anthropogenic activities. Such occurrence is avoided by carbonization of agricultural waste, producing biochar [42]. Biochar is the product of the thermochemical combustion of biomass (for example, corn cob, cocoa pod, and palm kernel husk) in a limited or no-oxygen environment [138]. Biochar has received increasing interest as an approach to address the challenges of global climate change and also to ensure sustainable agricultural development [139]. This material has been recently investigated to capture P and to recover wastewaters [140,141].

An emerging C-based material recently discovered is graphene [142], p. 20. It has attracted widespread attention for a broad range of applications including the use of energy-related materials [143], medication-administration systems [144], sensors [145,146], and membranes [147]. Graphene and its oxidized form, graphene oxide, has a range of reactive oxygen, functional groups, and a high specific surface. Furthermore, it has been confirmed by many studies as a non-toxic and biocompatible material [68,148].

On the basis of their high specific surface (2600 m$^2$ g$^{-1}$) and a unique 2D structure, graphene-based products present an ideal form to retain a high nutrient charge, due to the enormously negative charge on their surface. Other formulas using graphene oxide together with Fe are currently being tested, which would work as phosphate ion carrier. In this way, the supply of nutrients to plants can be improved while limiting leaching due to the slow P release [44].

## 9. Conclusions

Due to the economic and environmental impact that MAP, DAP, TSP, and SSP generate in the agricultural sector as fertilizers of soluble sources, the application of conventional fertilizers should receive increasing focus.

Phosphorus is a primary macronutrient that plays an important role as an energetic activator in crop productivity and quality. Nevertheless, attention should be placed on the inadequate use of chemically synthesized fertilizers for more economic and environmentally sustainable crop systems.

For proper agronomic decisions, several factors need to be taken into account, using alternative sources and recycling P from wastes and organic matrices in a more circular concept. A correct application of P in crop systems requires also an appropriate knowledge of the complex interaction between soil microbiota and plants and their interconnections with waterbodies and atmosphere to perform more accurate interpretations and recommendations at a local and regional level.

**Author Contributions:** Conceptualization, R.L.-T. and E.F.-O.; methodology, R.L.-T. and E.F.-O.; software, M.P.R.-M.; validation, R.L.-T., M.P.R.-M., L.C. and E.F.-O.; formal analysis, R.L.-T. and M.P.R.-M.; resources, L.C. and E.F.-O.; data curation, E.F.-O.; writing—original draft preparation, R.L.-T., M.P.R.-M., L.C. and E.F.-O.; writing—review and editing, R.L.-T., M.P.R.-M., L.C. and E.F.-O. All authors have read and agreed to the published version of the manuscript.

**Funding:** This research received no external funding.

**Acknowledgments:** Department of Soil Science and Agricultural Chemistry at the University of Granada (Spain), Department of Agricultural, Forest and Food Sciences (D.I.S.A.F.A. by its acronym in Italian) of University of Turin (Italy), and the authors appreciate the support of this work by Colfuturo and Minciencias scholarship in Colombia, Spanish Ministry of Economy and Competitiveness (Project CGL-2013-46665-R), and the European Regional Development Fund (ERDF).

**Conflicts of Interest:** The authors declare no conflict of interest.

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
