# Peer review of "Phosphorus Dynamics in the Soil–Plant–Environment Relationship in Cropping Systems: A Review"

_applsci, doi:10.3390/app112311133_

Round 1
Reviewer 1 Report
See specific comments:
Introduction:
L43: “Low P concentrations in the soil…” low just in available/labile forms.
The information in Figure 1 is not explained throughout the introduction section. For example, nitrogen is presented in figure 1, however, its relationship with P dynamics is poorly approached. I suggest authors separate the explanation about N from the Introduction (maybe creating an additional section focused on the content presented in figure 1).
Material and methods
Please provide the search terms used in the literature survey. And from which source they were extracted (title, keywords, abstract, etc.).
I strongly suggest authors elaborate a global map (new figure) including/identifying the origin of studies worldwide.
The original data from each study must be shown in tables (at least as supplementary material). Follow the example of table 1.
Section 5.0: Overall, this section is not fully linked to the information presented in figure 2. This section needs to be reformulated entirely to find a harmonious relationship between P and microorganisms.
Please replace “soil microbiology” with “soil microorganisms” or “soil microbiota”
Section 6.0: there is a low relationship between what is written in this section with the information available in figure 3. I strongly suggest authors insert data from studies reported in section 6 into figure 3. This will enrich the information shown in the figure. For instance, insert values (g kg-1 and %) to clarify the P dynamics in the soil-plant-environment system.
Alternative sources of P need to be presented in this Review. I suggest authors insert a new section on this focusing before the Conclusions section.
Reviewer 2 Report
Overall, an attention raising review with lots of facts and knowledge about the relationship between phosphorus and agriculture. However, it would have been better to focus on a narrower topic and elaborate it deeper and more detailed. Some topics are discussed in too much detail, while others are only mentioned without any substantive evaluation. There are unnecessary repetitions (P. 4 l. 156-157, P. 8 l. 320-321, P. 11 l. 382-383).
It describes the efficiency of phosphorus fertilization and environmental aspects, but does not shed enough light on the role of soil types, cultivated plant species or the advisory systems (extension services).
Specific comments:
- 1 l. 13: Unfortunately, not too many specific GAPs are described in the manuscript.
- 1 l. 20: “nuclear energy” – The rest of the manuscript has not a word about nuclear energy.
- 1 l. 20-21: “Climate change accounts for the availability of this element depending on whether the component is labile or non-labile” – strange wording, please rephrase
- 1 l. 37: “The P dynamics start with limitations and losses” - Strange wording, please rephrase.
- 1 l. 44: “this practice has not been demonstrated to be effective in a general way” – Please specify. The reader may ask, if not effective, why phosphorus fertilizers are used worldwide.
- 2 l. 61: “adequate range of 0.1-0.3 g kg-1” - Please clarify exactly what this value refers to. If it refers to the soluble P content of the soil, the extractant must also be described and a reference required.
- 3 l. 89: “Starting when? When will we apply results from other years?” – Unnecessary sentences
- 3 l. 116-117: “this element has a high solubility rate” Please explain this statement in more detail. Which form of P has high solubility rate in which environment or dissolvent?
- 4 l. 123: delete: “(Gustafsson et al., 2012)”
- 4 l. 139: “in some environments such as agricultural soils” – Agricultural soils are the typical environment for crop production. Please rephrase the sentence.
- 4 l. 149: Please specify. Which waters are not sensitive to P concentrations?
- 4 l. 162: The word “respectively” is difficult to interpret. One might think that 560 applies to Europe and 1115 to Oceania. Also change the point to comma: 1,115
- 5 l. 182: “improving the soil P absorption in monocotyledons and dicotyledons” – Please rephrase the sentence
- 8 l. 278-281: “Authors [101] determined that the response (productivity) of CO2 on low P concentrations (under 1,8 g kg-1 279 in the leaf) were higher than with greater concentrations; This was also the case with crops such as cotton and wheat (grass), which performed the same way.” – This sentence makes no sense. Please rephrase.
- 8 l. 284: Instead of “phonologic” maybe “phenologic(al)”?
- 8 l. 305-307: not “ml L-1”, but “µl L-1” or “ppm”
- 8 l. 312-313: Here, the causes and responses are not clear. Plant growth is always limited by the minimum nutrient or environmental factor. Plant growth does not require higher nutrient demand. Please rephrase the sentence.
- 8 l. 315-316: “a small concentration of the total P in the soil is found in form of labile phosphatized ions” – Please specify this small concentration (roughly).
- 10 l. 357-358: “its absorption from the soil by the plant is slow over short distances of approx. 0.1 to 0.15 mm” – How slow is it? Please specify.
- 12 l. 420-421: please explain in more detail the results and conclusions of the biochar experiments
Reviewer 3 Report
In the abstract this sentence:" Highlight relevant aspects of agronomic dynamics of phosphorus in the soil-plant relationship as a community (crop ecophysiology), the effect of environmental conditions and global warming on the redistribution and translocation of P concentrations in some commercial crops, and the use of Good Agricultural Practices (GAP) with aim to improve the efficiency of the element" needs a verb to be completed.
Through the manuscript, you should unify either the chemical name of elements or the symbols, for example sometimes you write nitrogen, phosphorus,….. and sometimes you write N, P,…
In line 52-54: " Hence phosphatized fertilization (due to the low mobility of this element in the soil) must be applied …………. half or full circle" please add ref..
In lines 277 and 278: the sentence " On the other hand, the high CO2 emission under P deficiency depressed productivity variables such as biomass or rice efficiency" needs ref.
The review was linked to specific places with different climates, in addition to the fact that within the review, no distinction was made or clarification of the difference between the dynamics of phosphorous in these places, so it is preferable to generalize and not link
Round 2
Reviewer 1 Report
The authors have done an excellent job. The article deserves to be published in its current form.